# Allele Frequencies and Forensic Data of 25 STR Markers for Individuals in Northeast Brazil

**DOI:** 10.3390/genes14061185

**Published:** 2023-05-29

**Authors:** Natalia Bahia Pinheiro dos Santos, Márcio Fabrício Falcão de Paula Filho, Abigail Marcelino dos Santos Silva, Enio Paulo Teló, José Bandeira do Nascimento Junior, Valdir de Queiroz Balbino, Iukary Oliveira Takenami, Isaac Farias Cansanção

**Affiliations:** 1Escola de Ciências da Saúde, Universidade Salvador, Salvador 41720-200, BA, Brazil; 2Laboratório de Investigação do Vínculo Genético, Centro de Diagnóstico do GACC (CDG), Grupo de Apoio à Criança com Câncer (GACC), Salvador 41250-010, BA, Brazil; 3Medicine Collegiate, Campus Paulo Afonso, Federal University of San Francisco Valley (UNIVASF), Amizade Avenue, 1900, Paulo Afonso 48605-780, BA, Brazil; 4Laboratory of Bioinformatics and Evolutionary Biology, Department of Genetics, Federal University of Pernambuco, Recife 50740-580, PE, Brazilbandeiraajr@gmail.com (J.B.d.N.J.);

**Keywords:** medical and population genetics, statistical genetics and genomics, short tandem repeats, allele frequencies, Brazilian population

## Abstract

Identifying DNA markers such as Short Tandem Repeats (STR) can be used to investigate genetic diversity based on levels of heterozygosity within and between populations. Allele frequencies and forensic data for STRs were obtained from a sample of 384 unrelated individuals living in Bahia, Northeastern Brazil. Thus, the present study aimed to identify the allele frequency distribution, in addition to the forensic and genetic data, of 25 STR loci in the population of Bahia. Buccal swabs or fingertip punctures were utilized to amplify and detect 25 DNA markers. The most polymorphic loci were SE33 (43), D21S11, and FGA (21). The least polymorphic were TH01 (6), TPOX, and D3S1358 (7). Forensic and statistical data were obtained through data analysis, which revealed a large genetic diversity, with an average value of 0.813 for the analyzed population. The present study was more robust than previous STR marker studies and will contribute to future research on population genetics in Brazil and worldwide. The results of this study allowed the establishment of haplotypes found in the forensic samples of Bahia State to serve as a reference in the elucidation of criminal cases and paternity tests, as well as population and evolutionary investigations.

## 1. Introduction

Advances in molecular biology have provided powerful tools for reconstructing genetic history. DNA contains valuable information that includes sequences from the evolutionary past that can be extracted from any type of biological material [1,2]. Human DNA sequences are now being used to study how distinct cultural groups are genetically related [3].

The identification of DNA markers such as short tandem repeats (STR) or microsatellites, which constitute highly variable genetic loci, has become the test of choice for genetic linkage analysis [4]. The approach is based on the identification of repetitive human DNA regions, which are characterized by size variation. A microsatellite region or locus consists of one to six base-pair sequences repeated several times throughout the genome. These are co-dominant markers and therefore can be used to investigate genetic diversity based on levels of heterozygosity within and between populations [5].

Moreover, the identification of polymorphisms is of great importance for the reconstruction of historical human migrations [5]. Since its discovery, Brazil has been a major focus of immigration. The Brazilian population is derived from the interbreeding of Portuguese, Africans, Europeans, Japanese, Brazilian natives, and many other nationalities. In geographically separated populations, the allelic distribution of DNA markers is often different. In Bahia, the highly mixed-sex population is derived from Brazilian natives, from Atlantic West Africa, and from Europeans, but with a predominance of African ancestors as African slaves [6]. This diversity reveals an intense gene flow reflecting a peculiar pattern of geographic diversity, a very interesting scenario for genetic studies [7].

The use of bioinformatics tools provided statistical data with regard to population genetics through forensic parameters such as genetic diversity (GD), polymorphic information content (PIC), heterozygosity (H), and the probability of the Hardy-Weinberg equilibrium (*p*-value) [3,5,7]. These parameters play a fundamental role in population genetics by identifying similarities and differences between and within populations [7].

Forensic analyses and studies of the distribution of allele frequencies, GD, and PIC in heterozygous populations are necessary to create databases for reference populations and to obtain information on the genetics of the population under study. To determine the genetic diversity and gene flow in this population, the present study determined the distribution of allele frequencies in addition to forensic and genetic data from 25 STR loci in the Bahia population.

## 2. Materials and Methods

### 2.1. Samples and DNA Extraction

A total of 384 samples from unrelated individuals living in Bahia, northeastern Brazil, were obtained as secondary data from paternity cases performed between 2016 and 2017 at the Laboratório de Investigação de Vínculo Genético do Centro de Diagnóstico do Grupo de Apoio à Criança com Câncer (CDG). Patients had previously given consent to the paternity tests. DNA was extracted from buccal swabs or directly from fingertip puncture using FTA^®^ cards (Whatman™ Bioscience, Cambridge, UK) according to the manufacturer’s protocol [8]. Samples were quantified by real-time PCR to assess the quantity and quality of selected DNA. The isolated DNA was stored at 4 °C until amplification. The study was approved by the Research Ethics Committee of the Universidade Salvador (UNIFACS): CAAE: 68946617.9.0000.5033.

### 2.2. PCR Amplification and Fragment Analysis

DNA extract (5 µL) was used for amplification of the 25 STRs examined in this study: (D12S391, D2S441, D22S1045, D10S1248, D8S1179, D21S11, D3S1358, TH01, D16S539, D2S1338, VWA31, D18S51, D19S433, FGA, CSF1PO, TPOX, D5S818, D13S317, D7S820, SE33, D1S1656, CD4, D8S639, PENTA E, and PENTA D). Electrophoresis and genotyping were performed using an ABI PRISM 3500 Genetic Analyzer (Applied Biosystems, Foster City, CA, USA). Data collection was performed using ABI PRISM 3500-Data Collection v1.0 software (Applied Biosystems), and for profile analysis we used GeneMapper ID -X v1.2 software (Applied Biosystems).

### 2.3. Statistical Analysis

Samples were calculated to obtain statistical data and forensic parameters from 384 individuals from the state of Bahia. After obtaining the allele profiles, the allele frequencies of the 25 markers were determined. The allele frequencies were determined using the relative frequency method and the number of allele repeats observed in the samples. The data were analyzed using GenAlEx version 6.5 software [9] to calculate both forensic and statistical parameters, including: the number of alleles (Nall), observed heterozygosity (Ho), expected heterozygosity (He), total heterozygosity or genetic diversity (GD), polymorphic information content (PIC), match probability or identity probability (PI), exclusion probability (PE), and probability using the Hardy-Weinberg equilibrium (*p*-value).

## 3. Results

The allele frequencies for the 25 STR loci studied in Bahia’s population are presented in Table 1. A total of 347 alleles were detected, with 51 of those qualifying as rare alleles (Allele Frequency < 0.005). The highest number of rare alleles were found in SE33 (six rare alleles), which also consisted of the highest total number of alleles.

For each analyzed locus, Appendix A presents the range of alleles, the number of alleles obtained in each locus, the allele frequency, and the gender distribution identified by the locus AMEL. The most polymorphic loci were SE33 (43 alleles), D21S11 (21 alleles), and FGA (21 alleles). TH01 (six alleles) had the least polymorphic loci. The allele frequencies ranged from 0.0013 to 0.39583 (Appendix A).

Table 2 presents comparisons between the most frequent alleles in Bahia, (northeastern Brazil), and in two studies in different periods in Brazil and Portugal. Of the 16 loci in common between Bahia and Portugal (D3S1358, VWA31, D16S539, CSF1PO, TPOX, D8S1179, D21S11, D18S51, TH01, FGA, D5S818, D13S317, D7S820, SE33, PENTA D, and PENTA E), only six (CSF1PO, TPOX, D21S11, D5S818, D7S820, and PENTA E) presented the same result for the analyzed parameter, demonstrating both the distances between the Portuguese and Brazilian populations, and the formative relationships within the total population.

The forensic parameters obtained for the 25 loci are presented in Table 3. The values for He and Ho ranged, from 0.731 (TPOX) to 0.934 (SE33), and from 0.685 (TPOX) to 0.971 (SE33), respectively. The GD values ranged from 0.7316 (TPOX) to 0.9339 (SE33), with an average of 0.813. The PIC, PI, and PE values ranged from 0.6905 (TPOX) to 0.93 (SE33), 0.008 (SE33) to 0.113 (TH01), and 0.771 (TH01) to 0.989 (SE33), respectively. The *p*-values obtained ranged from below 0.001 (PENTA D) to 0.9998 (FGA). No deviations from the Hardy-Weinberg equilibrium were observed. The high values obtained for these forensic parameters in the present study confirms the high genetic variability of Bahia’s population.

## 4. Discussion

This is the first study in Brazil reporting on 25 markers that were analyzed to evaluate the genetic variability of a specific human population. The allelic frequencies were compared to other regions of Brazil. No relevant differences were observed when comparing the frequencies of other studies performed in Rio Grande do Norte, Paraíba, Pernambuco, Santa Catarina, Mato Grosso do Sul, Rio Grande do Sul, Rio de Janeiro, Amazonas, or Paraná, with one other Brazilian study (Brazil) [10,11,12,13,14,15,16,17,18,19,20]. No comparisons were made for CD4, D8S639, and PENTA D, due to the lack of studies analyzing these markers.

Among observed studies, the most polymorphic loci were D2S1338, D18S51, D21S11, PENTA E, FGA, and SE33. In all studies, the least polymorphic locus was TPOX [10,11,12,13,14,15,16,17,18,19,20]. This study of Bahia’s population identified the SE33, D21S11, and FGA loci as having the highest polymorphism, with SE33 being found only in the Portuguese, Brazilian, and Northeastern Brazil studies [21].

The least polymorphic loci in the study with the Bahia population were TH01 and TPOX, similar to the other studies [10,11,12,13,14,15,16,17,18,19,20]. We also observed some rare alleles that were found only in this study, such as 28.1 and 30.3 of the D21S11 locus; 9.1 and 13.1 of the D7S820 locus; 19 of the D8S1179 locus; 4 of the D19S433 locus; 7.3 and 21.3 of locus SE33; and 13.3 of locus D1S1656.

We suggest that the movement of individuals from one population to another (immigration and emigration) implies the identified gene flow. In Northeastern Brazil, migrants reproduced and, via their genes, contributed to the genetic total of the receiver population [22], with the greatest contribution being the African component. There was little contribution from Native Americans. This is demonstrated when comparing the genetic frameworks between regions [19].

Table 2 compared the studies of Bahia, Northeastern Brazil, Brazil, and Portugal for the most frequent alleles of each analyzed locus. Of the 16 loci in common between Bahia and Portugal (D3S1358, VWA31, D16S539, CSF1PO, TPOX, D8S1179, D21S11, D18S51, TH01, FGA, D5S818, D13S317, D7S820, SE33, PENTA D, and PENTA E), only six (CSF1PO, TPOX, D21S11, D5S818, D7S820, and PENTA E) presented the same result for the analyzed parameter, demonstrating both distances between the Portuguese and Brazilian populations, and the formative relationships within the total population.

Few studies have contemplated the 25 markers presented in this study in Northeastern Brazil. When we compared the most frequent alleles for each locus of a study of the Brazilian population and that of Bahia, we noticed that among 13 loci analyzed (D3S1358, VWA31, FGA, D8S1179, S21S11, D18S51, D5S818, D13S317, D7S820, CSF1PO, TPOX, TH01, and D16S539), six presented similarities (D16S539, CSF1PO, TPOX, D21S11, D5S818, and D7S820). In another study, we noticed that of 21 loci studied (D3S1358, VWA31, D16S539, CSF1PO, TPOX, D8S1179, D21S11, D18S51, D2S441, D19S433, TH01, FGA, D22S1045, D5S818, D13S317, D7S820, D10S1248, D1S1656, D12S391, D2S1338, and SE33), 17 presented similarities (VWA31, D16S539, CSF1PO, TPOX, D21S11, D18S51, D2S441, D19S433, TH01, D5S818, D13S317, D7S820, SE33, D10S1248, D1S1656, D12S391, and D2S1338). According to Moyses et al. [19], this demonstrates the approximation between the genetic profiles of the populations between the Brazilian studies and Bahia, as well as for studies of Northeastern Brazil and Bahia.

In addition to allelic analysis, the present study provides statistical and forensic data for the 25 analyzed STR loci in Bahia’s population. Other Brazilian studies realized in this category present an average of 8 to 13 analyzed loci [10,11,12,13,14,15,16,17,18,19,20]. The present study thus promotes greater quantitative certainty for the genetic data of the Brazilian population.

The analysis of Genetic Diversity (GD) values ranged from 0.7316 (TPOX) to 0.9339 (SE33) (Table 3). These high values reveal the high genetic variability of Bahia’s population since the heterozygosity values approached their maximum values [22].

The expected heterozygosity (He) was higher than or equal to the observed heterozygosity (Ho) in 14 of the 25 loci analyzed (CD4, PENTA E, PENTA D, D3S1358, VWA31, D16S539, TPOX, D8S1179, D18S51, D22S1045, D13S317, D7S820, D10S1248, and D12S391). The highest discrepancy between parameters was 0.046 (TPOX). He values ranged from 0.731 (TPOX) to 0.934 (SE33), while Ho values ranged from 0.685 (TPOX) to 0.971 (SE33). The higher Ho as compared to He in the other 11 loci (D8S639, CSF1PO, D21S11, DS2441, D19S433, TH01, FGA, D5S818, SE33, and D1S1656, D2S1338) is indicative of external breeding [22] since the samples were obtained at random. The range obtained for the Ho and He values reveals that Bahia’s population presents a high genetic content.

Previous small-scale forensic studies in other northeastern states, such as Rio Grande do Norte, Paraíba, and Pernambuco have shown similar values for the discrepancy between observed and expected heterozygosity [10,11,12]. Due to the lack of studies, and on larger scales, it is not possible to establish average values for the forensic parameters analyzed across the Brazilian territory.

However, when comparing studies in other states such as Santa Catarina, Rio Grande do Sul, Rio de Janeiro, Mato Grosso do Sul, and Amazonas, it is possible to identify a preliminary average discrepancy between Ho and He which is lower than 0.05 for the Brazilian population [13,14,15,16,17], revealing a high genetic variation in the studied populations.

The same type of study done on a small scale with other populations and regions worldwide shows that populations with a history of miscegenation, such as Brazil, Africa, and northern Europe, present substantial genetic diversity and low discrepancies between He and Ho. Populations with low miscegenation present low heterozygosity [23,24]. A study done in Guangxi Zhuang, China, revealed a high discrepancy between these parameters (0.4975). This was attributed to the lack of geographic spreading and breeding among the native populations [23]. As highlighted in the present study, Bahia’s population was revealed to be a genetically varied population with a history of high miscegenation.

The values for PIC obtained in this study were higher than 0.6 for every locus analyzed, ranging from 0.6905 (TPOX) to 0.93 (SE33), which means that each of the 25 loci analyzed were highly polymorphic, and would contribute to the genetic variation of the analyzed population. Previous studies in other populations realized with PIC have shown similar results within some of the analyzed loci: TPOX, CSF1PO, PENTA E, FGA, and TH01 [11,12,13]. Due to the number of loci analyzed in the present study, further comparisons were not possible with the published studies. This reveals the importance and necessity of completing more studies on the loci in the miscegenated population analysis, especially in Brazil.

Values for probability of identity (PI) within the studied loci ranged from 0.008 (SE33) to 0.113 (TH01). TH01, DS2441, TPOX, and CSF1PO presented the highest PI values in our analysis, ranging from 0.1046 (CSF1PO) to 0.113 (TH01). Previously available studies on TH01, TPOX, and CSF1PO in northeastern Brazil have shown similarities in values for PI in these same loci [10,11,12]. The PI range must fall between 0 and 1 [9], and thus, given the values obtained, every locus analyzed in this study presented a considerable difference when comparing genotypes in the studied population, confirming the high genetic variability of Bahia’s population.

For the analyzed group, the PE ranged from 0.771 (TH01) to 0.989 (SE33) for the loci studied. The PE values were higher than 0.7 for every locus, confirming a great genetic variability in the genetic profile of Bahia’s population. Previous studies undertaken in northeastern Brazil with PE analysis demonstrate only three loci in common with the present study (TH01, TPOX, and CSF1PO) [10,11,12]. The PE values obtained from these loci are evidence that Bahia’s population maintains a higher genetic variation than other populations in the region. The PE averages for these same loci in the other studies were 0.5713 (TH01), 0.4223 (TPOX), and 0.4876 (CSF1PO).

The *p*-value obtained in this study ranged from below 0.001 (PENTA D) to 0.9998 (FGA). A total of 18 of the 25 analyzed loci (CD4, PENTA E, D3S1358, D16S539, CSF1PO, D21S11, D18S51, DS2441, D19S433, TH01, FGA, D5S818, D13S317, D7S820, SE33, D1S1656, D12S391, and D2S1338) presented non-significant *p*-values (*p* > 0.05). Seven of them (D8S639, PENTA D, VWA31, TPOX, D8S1179, D22S1045, and D10S1248) presented significant *p*-values (*p* < 0.05). Thus, the vast majority of the loci in the analyzed population either reached or were likely to reach the Hardy-Weinberg equilibrium [22,25].

Similar results for the CSF1PO, TH01, TPOX, VWA31, D16S539, D13S317, D21S11, D8S1179, and FGA loci have been found in studies conducted in the Rio Grande do Norte, Paraíba, and Pernambuco [10,11,12]. These studies, which were performed in geographically confined populations, presented *p*-values approximating *p* > 0.05, except for the TPOX locus, which presented a *p*-value of < 0.05 in Bahia’s population. This scenario reveals the genetic proximity of the studied populations, which can be explained by their geography and history.

## 5. Conclusions

When compared to other populations, an analysis of the data confirms that Bahia’s population is genetically diverse. Furthermore, the present study provides statistical forensic data that may help guide future research on STR markers.

When comparing the allele frequency of the 25 STR markers in Bahia’s population with other studies done in different populations in northeastern Brazil, no significant differences were found, revealing great similarities between populations. When forensically analyzed, Bahia’s population reveals great genetic variety, a sign of significant miscegenation in its genetic formation. The allelic frequencies of the studied population revealed contributions from Native Americans, Africans, and the Portuguese. The results also show that studies in specific populations are more likely to produce reliable results. To better understand the genetic diversity of this particular Brazilian population, more studies on the populations that contributed to its formation are needed.

Finally, the present study provided forensic and statistical data that may help guide future research on STR markers and forensic studies in miscegenated populations, such as that of Brazil.

## Figures and Tables

**Table 1 genes-14-01185-t001:** Allele frequency distribution and gene diversity in Bahia, Brazil.

Allele	CD4	*D8S639*	*PENTA E*	*PENTA D*	*D3S1358*	*VWA31*	*D16S539*	*CSF1PO*	*TPOX*	*D8S1179*	*D21S11*	*D18S51*	*D2S441*	Allele
2.2			0.001	0.055										2.2
3.2				0.004										3.2
4														4
5	0.328		0.069	0.023			0.001							5
6	0.190			0.001				0.001	0.038					6
7	0.007		0.122	0.013				0.022	0.004					7
7.3														7.3
8	0.061		0.104	0.060			0.021	0.038	0.396	0.004				8
9	0.014		0.018	0.164			0.165	0.029	0.163	0.008		0.001	0.003	9
9.1														9.1
9.2				0.005										9.2
9.3														9.3
10	0.206		0.057	0.171			0.102	0.273	0.077	0.044		0.009	0.195	10
10.2												0.003		10.2
10.4	0.001													10.4
11	0.105	0.001	0.109	0.158		0.008	0.319	0.263	0.276	0.072		0.005	0.324	11
11.2														11.2
11.3													0.059	11.3
12	0.057		0.132	0.138			0.236	0.319	0.047	0.121		0.092	0.099	12
12.2														12.2
12.3													0.005	12.3
13	0.013		0.121	0.133	0.007	0.009	0.130	0.043		0.255		0.091	0.029	13
13.1														13.1
13.2												0.001		13.2
13.3														13.3
14	0.010		0.051	0.047	0.107	0.069	0.022	0.012		0.279		0.151	0.253	14
14.2														14.2
14.3														14.3
15	0.007		0.070	0.026	0.280	0.172	0.004			0.160		0.158	0.033	15
15.2														15.2
15.3														15.3
16		0.001	0.052		0.290	0.253				0.052		0.154	0.001	16
16.2														16.2
16.3														16.3
17			0.029		0.208	0.233				0.003		0.109		17
17.2														17.2
17.3														17.3
18			0.023		0.103	0.172				0.001		0.094		18
18.2														18.2
18.3														18.3
19			0.017		0.005	0.066				0.001		0.076		19
19.1														19.1
19.2														19.2
19.3														19.3
20			0.014			0.017						0.034		20
20.2														20.2
21		0.001	0.004			0.001						0.010		21
21.2												0.001		21.2
21.3														21.3
22		0.008	0.004									0.007		22
22.2														22.2
23		0.003	0.001									0.003		23
23.2														23.2
24		0.025		0.001										24
24.2														24.2
24.3											0.004			24.3
25		0.081										0.001		25
25.2											0.001			25.2
26		0.121		0.001							0.001			26
26.2														26.2
27		0.208									0.033			27
27.2														27.2
27.3		0.003												27.3
28		0.191									0.173			28
28.1											0.001			28.1
28.2														28.2
28.3		0.009												28.3
29		0.105									0.201			29
29.2														29.2
29.3		0.056												29.3
30											0.233			30
30.2											0.026			30.2
30.3		0.042									0.001			30.3
31											0.060			31
31.2											0.089			31.2
31.3		0.051												31.3
32											0.021			32
32.2											0.089			32.2
32.3		0.057												32.3
33											0.001			33
33.2		0.029									0.022			33.2
34											0.007			34
34.2											0.004			34.2
34.3		0.008												34.3
35											0.023			35
36											0.009			36
37											0.001			37
**Allele**	** *D19S433* **	** *TH01* **	** *FGA* **	** *D22S1045* **	** *D5S818* **	** *D13S317* **	** *D7S820* **	** *SE33* **	** *D10S1248* **	** *D1S1656* **	** *D12S391* **	** *D2S1338* **		**Allele**
2.2														2.2
3.2														3.2
4	0.001													4
5														5
6		0.199												6
7		0.309			0.012		0.010							7
7.3								0.001						7.3
8		0.176			0.035	0.055	0.189							8
9		0.143			0.033	0.073	0.103	0.001	0.001					9
9.1							0.001							9.1
9.2								0.001						9.2
9.3		0.161												9.3
10	0.008	0.012		0.021	0.072	0.033	0.268		0.004	0.012				10
10.2								0.003						10.2
10.4														10.4
11	0.044			0.117	0.315	0.302	0.234	0.001	0.018	0.061				11
11.2								0.001						11.2
11.3														11.3
12	0.105			0.030	0.322	0.326	0.156	0.001	0.090	0.077				12
12.2	0.036							0.007						12.2
12.3														12.3
13	0.251			0.005	0.185	0.161	0.035	0.007	0.271	0.099				13
13.1							0.001							13.1
13.2	0.046							0.007						13.2
13.3										0.001				13.3
14	0.240			0.053	0.025	0.048	0.001	0.036	0.306	0.167		0.001		14
14.2	0.040							0.003						14.2
14.3										0.013				14.3
15	0.125			0.303	0.003	0.003		0.049	0.181	0.172	0.055			15
15.2	0.047							0.001						15.2
15.3										0.042				15.3
16	0.036		0.001	0.307				0.069	0.104	0.111	0.047	0.046		16
16.2	0.014													16.2
16.3								0.003		0.065				16.3
17	0.001		0.001	0.155				0.091	0.023	0.046	0.128	0.174		17
17.2	0.004													17.2
17.3								0.001		0.100	0.005			17.3
18			0.003	0.007				0.109	0.001	0.003	0.224	0.070		18
18.2			0.004											18.2
18.3										0.030	0.012			18.3
19			0.086	0.001				0.107			0.158	0.159		19
19.1											0.003			19.1
19.2			0.003											19.2
19.3										0.003	0.005			19.3
20			0.083					0.074			0.150	0.122		20
20.2								0.009						20.2
21			0.161					0.042			0.073	0.065		21
21.2								0.018						21.2
21.3								0.001						21.3
22			0.148					0.016			0.064	0.095		22
22.2			0.003					0.010						22.2
23			0.167					0.004			0.048	0.086		23
23.2			0.003					0.012						23.2
24			0.159								0.020	0.076		24
24.2								0.026						24.2
24.3														24.3
25			0.109					0.001			0.007	0.069		25
25.2			0.001					0.038						25.2
26			0.038					0.001			0.004	0.030		26
26.2								0.057						26.2
27			0.013									0.007		27
27.2								0.056						27.2
27.3														27.3
28			0.009											28
28.1														28.1
28.2								0.052						28.2
28.3														28.3
29			0.003											29
29.2								0.025						29.2
29.3														29.3
30														30
30.2			0.001					0.023						30.2
30.3														30.3
31														31
31.2			0.004					0.018						31.2
31.3														31.3
32														32
32.2								0.004						32.2
32.3														32.3
33														33
33.2														33.2
34								0.005						34
34.2								0.005						34.2
34.3														34.3
35								0.001						35
36														36
37														37

**Table 2 genes-14-01185-t002:** The most frequent alleles in Bahia, northeast, Brazil, and Portugal.

Locus	Bahia	Northeast_9_	Brazil_18_	Brazil_17_	Portugal_19_
D3S1358	16	16	15	15	15
VWA31	16	16	17	16	17
D16S539	11	11	11	11	12
CSF1PO	12	11	12	12	12
TPOX	8	8	8	8	8
D8S1179	14	13	13	13	13
D21S11	30	30	30	30	30
D18S51	15	15	16	15	16
D2S441	11	11	*	11	*
D19S433	13	13	*	13	*
TH01	7	7	9.3	7	9.3
FGA	23	24	22	24	22
D22S1045	16	15	*	15	*
D5S818	12	12	12	12	12
D13S317	12	12	11	12	11
D7S820	10	10	10	10	10
SE33	18	18	*	18	17
D10S1248	14	14	*	14	*
D1S1656	15	15	*	15	*
D12S391	18	18	*	18	*
D2S1338	17	17	*	17	*
PENTA D	10	*	*	*	13
PENTA E	12	*	*	*	12

* Data not related.

**Table 3 genes-14-01185-t003:** Data Analysis and forensic parameters obtained for Bahia’s population.

Allele	CD4	D8S639	PENTA E	PENTA D	D3S1358	VWA31	D16S539	CSF1PO	TPOX	D8S1179	D21S11	D18S51	D2S441	D19S433	TH01	FGA	D22S1045	D5S818	D13S317	D7S820	SE33	D10S1248	D1S1656	D12S391	D2S1338
N	384	384	384	384	384	384	384	384	384	384	384	384	384	384	384	384	384	384	384	384	384	384	384	384	384
Nall	11	16	19	15	7	10	9	9	7	12	14	16	8	9	5	16	10	9	8	8	27	10	10	12	13
Ho	0.773	0.872	0.904	0.831	0.734	0.781	0.784	0.768	0.685	0.805	0.846	0.865	0.753	0.818	0.779	0.883	0.760	0.766	0.750	0.781	0.971	0.776	0.885	0.844	0.898
He	0.795	0.859	0.910	0.871	0.772	0.813	0.787	0.750	0.731	0.807	0.825	0.884	0.739	0.779	0.741	0.868	0.772	0.755	0.765	0.801	0.934	0.780	0.849	0.860	0.890
GD	0.7949	0.8589	0.9097	0.8703	0.7719	0.813	0.7871	0.7498	0.7316	0.807	0.8255	0.88377	0.7387	0.779	0.7412	0.8685	0.7714	0.7559	0.7649	0.8008	0.9339	0.7804	0.849	0.8597	0.8903
PIC	0.7677	0.8435	0.9027	0.8564	0.7352	0.7868	0.7563	0.7079	0.6905	0.7809	0.8028	0.8723	0.6976	0.7454	0.6944	0.8542	0.7373	0.7178	0.73	0.7713	0.93	0.7476	0.8309	0.8447	0.8802
PI	0.069	0.035	0.015	0.031	0.089	0.061	0.076	0.105	0.114	0.063	0.053	0.025	0.109	0.082	0.114	0.032	0.086	0.098	0.090	0.069	0.008	0.081	0.041	0.035	0.022
PE	0.943	0.972	0.980	0.968	0.871	0.932	0.917	0.917	0.871	0.951	0.963	0.972	0.898	0.917	0.772	0.972	0.932	0.917	0.898	0.898	0.990	0.932	0.932	0.951	0.958
*p*-value	0.199	0.000	0.984	0.000	0.632	0.035	0.375	0.879	0.012	0.000	0.166	0.189	0.878	0.475	0.467	0.9998	0.001	0.762	0.808	0.050	0.881	0.0003	0.341	0.054	0.993

N: Number of individuals; Nall: Number of alleles; Ho: Observed heterozygosity; He: Expected heterozygosity; GD: Genetic diversity (total heterozygosity); PIC: Polymorphic information content; PI: Probability of identity; PE: Probability of exclusion; *p*-value: Probability of Hardy-Weinberg equilibrium.

## Data Availability

The data used to support this study are included in this paper and as Appendix A.

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
