# Peer review of "Allele Frequencies and Forensic Data of 25 STR Markers for Individuals in Northeast Brazil"

_genes, 2023, doi:10.3390/genes14061185_

Round 1

Reviewer 1 Report

Dear authors,

The article you submit , shows the distribution of allele used in forensic sciences and shows the frequencies for individuals in Northern Brazil.

This article is clearly presented, and the study is well explained.

It could be discussed more extensively, "in Statistical analysis" the number of samples “384”, is it sufficient and/or representative of the population., Which percentage of the population.

The presentation of table 1 and table 2, must be improved with the presentation, these two tables with all the data are very interesting, but it is very difficult to read from one page to the next one. For example, the format of presentation the table 4, is more clear.

In the conclusion, an extension could be given for the use of these data in a real forensic context, possibly this article could be submitted to MDPI " Forensic Sciences"

Kind regards

Author Response

Dear reviewer,

Our answers are attached.

Kind regards,

Reviewer 2 Report

Even if the number of individuals tested for this population study is lower (current requirement for human population genetics is >500), the importance of the data for the forensic community and subsequently the law enforcement agencies is significant.

Author Response

(The authors gave the same response as above.)

Reviewer 3 Report

Overall, the work is useful as it deals directly with something that is, after all, the centre of forensic genetics: statistics and the study of frequencies. Some points deserve improvement, as explained below:

1. The first paragraph of the introduction is too abstract and general for such a definite article. The introduction should serve to explain the theoretical concepts necessary to understand the discussion and the parameters described therein.

2. Point 2.1

-It is not mentioned whether the individuals signed an informed consent, authorizing their biological samples to be used for this purpose.

- It is not explained what the extraction method was.

- Have the samples been quantified?

Author Response

(The authors gave the same response as above.)

Round 2

Reviewer 3 Report

Dear Authors,

The article is greatly improved and with greater scientific rigor.

My issue remains informed consent. The study itself has passed an ethics committee, meaning it can be done. On the other hand, the authors report that the individuals gave their consent for the paternity tests to be carried out. And this is where my problem lies, as these people gave the sample for a paternity test, not to be used in this study. Unless the ethics committee authorizes the use of samples without express consent for that purpose (i.e. expressly states that samples collected from paternities may be used for this investigation without the individuals knowing), these samples were used with a purpose other than the one for which they were collected.

Author Response

Question: Dear Authors,
The article is greatly improved and with greater scientific rigor.
My issue remains informed consent. The study itself has passed an ethics committee, meaning it can be done. On the other hand, the authors report that the individuals gave their consent for the paternity tests to be carried out. And this is where my problem lies, as these people gave the sample for a paternity test, not to be used in this study. Unless the ethics committee authorizes the use of samples without express consent for that purpose (i.e. expressly states that samples collected from paternities may be used for this investigation without the individuals knowing), these samples were used with a purpose other than the one for which they were collected.

Answer - Dear reviewer, thank you for the comments and your questions. However, we could not satisfactorily explain the consent of the participants in this study. The research was followed up and analyzed from different points of view, such as paternity testing, population frequency, and genetic linkage studies. One of these studies (like the research that led to this manuscript) was to analyze data that would allow for greater reliability in the study of genetic linkage. I emphasize that all participants were informed of these aspects at the time of their informed consent. Since this was time-limited research, this part of the research (the genetic and forensic linkage study) was started after the collection of the participants' data and the DNA was isolated later. Therefore, we reiterate our ethical stance and respect for the Declaration of Helsinki and the Brazilian Ethics and Research Commission.
